# Central Nervous System Progression in Primary Vitreoretinal Lymphoma with Bilateral and Unilateral Involvement: A Systematic Review and Meta-Analysis

**DOI:** 10.3390/cancers14122967

**Published:** 2022-06-16

**Authors:** Josephus L. M. van Rooij, Klaudia A. Tokarska, Ninette H. ten Dam-van Loon, Peter H. Wessels, Tatjana Seute, Monique C. Minnema, Tom J. Snijders

**Affiliations:** 1Department of Neurology and Neurosurgery, UMC Utrecht Brain Center, University Medical Center Utrecht, 3508 GA Utrecht, The Netherlands; j.l.m.vanrooij-7@umcutrecht.nl (J.L.M.v.R.); claudianna.tokarska@gmail.com (K.A.T.); t.seute@umcutrecht.nl (T.S.); 2Department of Ophthalmology, University Medical Center Utrecht, 3508 GA Utrecht, The Netherlands; n.tendam-vanloon@umcutrecht.nl; 3Department of Neurology, St Antonius Hospital Utrecht/Nieuwegein, 3430 EM Nieuwegein, The Netherlands; p.wessels@antoniusziekenhuis.nl; 4Department of Hematology, University Medical Center Utrecht, 3508 GA Utrecht, The Netherlands; m.c.minnema@umcutrecht.nl

**Keywords:** primary vitreoretinal lymphoma, primary intraocular lymphoma, primary central nervous system lymphoma, CNS involvement, neuro-oncology, ophthalmology

## Abstract

**Simple Summary:**

Primary vitreoretinal lymphoma (PVRL) is a rare disease with high mortality rates. It has a poor prognosis mainly because of its tendency to spread to the central nervous system (CNS). The optimal treatment strategy for PVRL is unknown; ideally, a treatment should prevent spread to the CNS, and thereby prolong overall survival. PVRL may occur in one eye (unilateral PVRL), or in both (bilateral). We reviewed studies from the scientific literature to investigate whether the risk of CNS progression differs between bilateral and unilateral PVRL. The quality of most available studies was moderate, at best. From the available studies, we found no difference in the development of CNS disease between patients with bilateral PVRL and unilateral PVRL.

**Abstract:**

Background: Primary vitreoretinal lymphoma (PVRL) is either unilateral or bilateral at initial presentation. Progression to a central nervous system (CNS) lymphoma is regularly observed and these patients seem to have an inferior survival. Knowledge of the predictive value of laterality for CNS progression may facilitate risk stratification and the development of more effective treatment strategies, and eventually, improve outcomes. The objective of this analysis is to estimate the risk of CNS progression for patients with bilateral versus unilateral involvement of PVRL. Methods: Systematic literature search for studies on CNS progression in PVRL with bilateral and unilateral involvement according to the Preferred Reporting Items for Systematic Reviews and Meta-Analyses (PRISMA) guidelines. We assessed the risk of bias and the methodological quality of studies using the Quality in Prognosis Studies (QUIPS) tool. Risk ratios of CNS progression in PVRL with bilateral and unilateral involvement were calculated and combined via a meta-analysis. Results: Twenty-five small-sized (total *n* = 371 cases) studies were included. The majority of the studies were at medium to high risk of bias. Results suggest no significant difference in CNS progression between bilateral and unilateral PVRL, with a pooled relative risk ratio of 1.12 (95% confidence interval 0.89–1.41). Conclusions: CNS progression is common in PVRL. From the limited available evidence, there is no significant difference in CNS progression between bilateral and unilateral PVRL.

## 1. Introduction

Primary vitreoretinal lymphoma (PVRL), also known as primary intraocular lymphoma (PIOL), is a rare form of non-Hodgkin lymphoma. Vitreoretinal lymphoma is defined as primary vitreoretinal lymphoma (PVRL) when there is no evidence of central nervous system (CNS) or systemic involvement at the time of diagnosis. It is, however, closely related to and defined as a subset of primary central nervous system lymphoma (PCNSL).

PVRL is a rare malignancy and typically affects elderly patients, with women appearing to be more commonly affected than men [1]. PVRL is either unilateral or bilateral at initial presentation, but it is estimated that approximately 80–90% of the patients will eventually develop bilateral disease [2].

When a diagnosis of PVRL is made, cerebrospinal fluid analysis and brain imaging are routinely performed to rule out concomitant CNS involvement, as well as imaging to rule out concomitant systemic involvement.

Approximately 15–25% of PCNSL patients present with vitreoretinal lymphoma [3]. Reports on the rate of CNS progression in PVRL vary widely, and risk factors for CNS progression are largely unknown [4,5]. The interval between the onset of PVRL and CNS progression was reported at 22 months but varies widely between patients [5]. Published mortality rates for PVRL vary widely between 9% and 81%, in follow-up periods ranging from 12 to 49 months [6,7]. The prognosis depends, to a large extent, on whether the CNS is involved. A trend toward better survival is observed among patients who are diagnosed with vitreoretinal lymphoma without concomitant CNS involvement at initial presentation [8].

Due to the rarity of the disease, the optimal treatment of PVRL is controversial and remains undefined. There are various treatments for PVRL, both local ocular therapy and systemic treatments. Studies report varying rates of effectiveness of these treatments, with reasonable ocular remission rates but high rates of recurrence and CNS progression [1,9,10]. Some experts believe that early systemic therapy for PVRL may prevent CNS progression, but a recent multicenter retrospective cohort study showed no difference in CNS progression between patients treated with local ocular treatment or systemic treatment [11]. The optimal treatment of PVRL should be designed not only to eradicate the ocular disease but also to prevent CNS progression.

Identification of those PVRL patients with the highest risk of CNS progression is important. High-risk patients may either receive early treatment for prevention of CNS progression–ideally in the setting of a prospective study—or be monitored intensively. It is unknown if the risk of CNS progression differs between bilateral and unilateral involvement in PVRL, and whether this clinical characteristic can be used for risk stratification. Hence, we performed a systematic and critical review of the literature on CNS progression in bilateral and unilateral PVRL in order to estimate the risk of CNS disease development for patients with PVRL with bilateral and unilateral involvement.

### Objectives

To perform a systematic and critical review of the literature on CNS progression in bilateral and unilateral PVRL to estimate the risk of CNS disease development for patients with PVRL: bilateral versus unilateral involvement.

## 2. Methods

This systematic review and meta-analysis were designed in 2020–2021 and undertaken in accordance with the PRISMA (Preferred Reporting Items for Systematic Reviews and Meta-Analyses) guidelines [12]. No ethical committee approval was required.

### 2.1. Search Strategy and Study Selection

We conducted a literature search for articles published until 30 December 2021, using international databases PubMed and Embase. The search strategy involved combining terms related to PVRL and CNS progression and is presented in Appendix A. In Embase (1805), limits were used with the exclusion of conference abstracts, editorial letters, and animal experiments. From the obtained papers, duplicates were removed. Two authors, JvR and KT, independently screened the records on title and abstract and assessed the records for eligibility. Disagreement on whether a study met the eligibility criteria was discussed with other authors (TJS and PW). After the search was conducted, we screened references of the included papers for potential additional papers that were not found based on our search strategy (‘snowballing’). These additional papers were first screened based on the title and then the full text.

### 2.2. Inclusion and Exclusion Criteria

All original studies on CNS involvement in PVRL conducted with human adult patients diagnosed with PVRL based on ophthalmic features were included in this review. Studies with the following criteria were excluded from this review: (1) studies with patients with recurrent PVRL; (2) studies with immunocompromised patients; (3) studies with patients with lymphoma localization outside the CNS; (4) studies written in languages other than English, Dutch, or Polish; (5) case reports or mini-series with fewer than five patients with PVRL; (6) manuscripts without original data, such as reviews, poster papers, letters, comments, and editorials. Conference abstracts were excluded as they contained insufficient data to perform meta-analysis and quality assessment.

A title and abstract screening selection based on these exclusion and inclusion criteria was first conducted for all selected studies, resulting in a collection of studies eligible for inclusion in the review. Data on laterality of PVRL and CNS progression incidence for bilateral and unilateral PVRL were extracted from these eligible studies. Authors were asked for this information by email in case of incomplete information, particularly for studies wherein information was available regarding CNS progression in general, but specific data on bilateral and unilateral PVRL were missing. Only the studies for which information on both the CNS progression and the laterality of PVRL was available, were included in this review.

### 2.3. Data Extraction

To extract data from the selected studies, we drafted a document analysis chart containing the following variables: authors, publication year, data collection site, inclusion period, study design, number of patients with PVRL, CNS progression of patients diagnosed with PVRL, laterality of PVRL, incidence of CNS progression in bilateral and unilateral PVRL, time to CNS progression, follow-up period of patients diagnosed with PVRL, and initial therapy strategies. Time to CNS progression is presented as a mean (with range) and the follow-up period as a median (with range) because of their tendencies towards a normal and skewed distribution, respectively. In some articles, the mean and median were stated in the text, but for most studies, we calculated the mean and median from the available data.

### 2.4. Risk of Bias Assessment

In order to assess the risk of bias of the included studies, the Quality in Prognosis Studies (QUIPS) checklist was used [13]. The purpose of this appraisal is to assess the methodological quality of a study and to determine the extent to which a study has addressed the possibility of bias in its prognostic factors. It contains six domains: study participation, study attrition, prognostic factor measurement, outcome measurement, study confounding, and statistical analysis and reporting (see Appendix A). The risk of bias was assessed independently by two authors, JvR and KT. Any disagreements were discussed with two other authors (TJS and PW). The ‘risk of bias’ assessment did not serve as a ground for exclusion from the final analysis.

### 2.5. Statistical Analysis/Meta-Analysis

The primary outcomes included: (1) the incidence of CNS progression in bilateral and unilateral PVRL, and (2) risk estimates, expressed as relative risk (RR) with a 95% confidence interval (CI). For statistical analysis, Review Manager Software version 5.4 (Nordic Cochrane Centre, Cochrane Collaboration) was used [14]. CNS progression incidence for bilateral PVRL compared to unilateral PVRL was reported and the relative risk (RR) was calculated from this data with 95% CI. If there were zeros in one of the groups, the value of 0.5 was added to all groups to calculate the relative risk. We pooled the relative risk outcomes of the studies in a meta-analysis. Studies describing patients with only unilateral or bilateral PVRL were excluded from the meta-analysis. Due to small study sizes with independently operating researchers, the Mantel–Haenszel method was used with a random-effects model [15]. The I^2^ statistic of Higgin and Thompson was used to quantify statistical heterogeneity [16]. Due to limited available information regarding different treatment groups for PVRL, no subgroup analysis was performed for different treatment groups. Furthermore, limited information was available regarding follow-up and time to event. It was not possible to explore possible causes of heterogeneity because of these missing data.

## 3. Results

### 3.1. Study Selection

An overview of the study selection process is shown in the flowchart following the PRISMA model in Figure 1. The literature search returned a total of 1415 unique studies. After exclusion of irrelevant articles based on title and abstract, mainly on the fact that there was a different study population, a full-text review was performed on 244 articles. Despite extensive efforts of the research department, eleven out of these 244 articles could not be retrieved. Of the remaining 233 articles, 211 articles were excluded because of the following reasons: (1) no or only limitedly information on CNS progression was stated; (2) the study population consisted mainly of patients with PCNSL, with less than five patients with PVRL; (3) no information on laterality of PVRL was given; (4) CNS progression rates were not given separately for bilateral and unilateral PVRL. There were initially 27 articles that did not contain information on CNS progression separately for bilateral and unilateral PVRL [5,11,17,18,19,20,21,22,23,24,25,26,27,28,29,30,31,32,33,34,35,36,37,38,39,40,41]. The corresponding authors were requested for this information. The authors of four studies [38,39,40,41] responded and provided us with the missing data, leading to the inclusion of these four studies. The remaining 23 studies were excluded from this review. Lastly, two records were identified from references and could be included in our study [42,43].

A final total of 25 articles met the eligibility criteria and were included in this systematic review [7,38,39,40,41,42,43,44,45,46,47,48,49,50,51,52,53,54,55,56,57,58,59,60,61].

### 3.2. Study Characteristics

The characteristics of the 25 included studies can be found in Table 1. We included eight studies from the USA [38,42,45,47,49,51,57,60], eight studies from Japan [7,48,52,53,54,58,59,61], five from Europe [41,43,46,50,56], and four from other Asian countries [39,40,44,55]. The studies included in this review were conducted in the 52-year period from 1968 to 2020. The majority of the studies were retrospective (76%). Five studies reported CNS progression as a primary outcome [7,38,52,54,61]. All the included studies were empirical studies and the sample size ranged from five to 59.

The included studies described a total of 371 patients initially diagnosed with PVRL, of whom 169 (46%) eventually developed CNS disease. Bilateral involvement was more common than unilateral involvement, with 67% of patients having bilateral involvement. CNS progression incidence ranged from 0% to 86% between studies. The mean time to CNS development ranged between eight and 34 months, with a range from 1–86 months. This information was missing from eight studies. The median follow-up period varied substantially between studies, with follow-up periods ranging from 1–166 months. Information on the follow-up period was not available for the two studies.

Treatment protocols used in the different studies ranged from only intravitreal methotrexate (IVM) in one study [55] to one study with many different complex treatment groups consisting of systemic chemotherapy, IVM, intravitreal rituximab (IVR), systemic rituximab, ocular radiation therapy, brain radiation therapy, and/or autologous stem cell transplantation [38]. For most of the studies, information on treatment for CNS progression was limited and was rarely reported separately for initial unilateral vs. bilateral ocular involvement.

Castellino et al. [38] described a subgroup analysis of patients with initial unilateral involvement: of the six examined patients, three received only ocular therapy (IVM and IVR) and three received systemic chemotherapy. CNS progression was solely described in one patient who was treated with only ocular therapy. Arai et al. [59] described the different treatment groups for unilateral and bilateral initial involvement: two out of three patients with initial bilateral involvement developed CNS progression, with all of them treated with a combination of systemic chemotherapy and IVM. Of the four patients who presented with unilateral involvement, two were treated with a combination of systemic chemotherapy and IVM and two with only IVM. Only one patient, who was treated with only IVM, developed CNS progression.

### 3.3. Risk of Bias

The risk of bias assessed using the QUIPS tool is shown in Table 2. A high risk of bias regarding study participation was reported in five studies since they described the baseline study sample and the inclusion and exclusion criteria incompletely and included patients with final diagnosis confirmation of PVRL after the detection of CNS progression (in retrospect). Fourteen studies lacked an adequate description of patients lost to follow-up and a comparison between patients who were lost to follow-up and those who were not. This resulted in a high risk of bias regarding attrition. Seven studies lacked a clear definition of the outcome and did not describe the method of CNS diagnosis. A high risk of bias for confounding variables was reported in five studies that did not adequately describe different treatments received by patients and did not take these into account when analyzing and reporting data. Eight studies had a moderate to high risk of bias for statistical analysis and reporting due to missing information on the relationship between laterality of PVRL and CNS progression. Overall, seven studies were assessed as having a high risk of bias, with the remaining eighteen articles being assessed as moderate risk of bias.

### 3.4. CNS Progression in PVRL

CNS progression rates for unilateral and bilateral involvement in PVRL are included in Table 3. The relative risk (RR) with 95% CI of CNS progression for bilateral PVRL compared to unilateral PVRL is also presented in Table 3. Four studies were excluded from the relative risk ratio analysis since all of their patients had bilateral PVRL [46,50,55,57]. The RR ranged from 0.38 to 3.82. The RR suggests a higher risk of CNS progression in patients with unilateral involvement (RR < 1) in four studies [41,44,53,61] and a higher risk of CNS progression in patients with bilateral involvement (RR > 1) in the remaining 17 studies. None of these RRs were significant, with large confidence intervals due to small sample sizes. In the two studies that included the most patients with PVRL, the non-significant RRs suggested a higher risk for unilateral involvement (RR < 1) [41,61].

### 3.5. Meta-Analysis

In order to combine the results, a meta-analysis was performed. Four studies were excluded based on the fact that all patients had bilateral PVRL [51,55,60,62]. The meta-analysis showed a pooled relative risk ratio of 1.12 (*p* = 0.33) A forest plot was created, showing the pooled relative risk with its 95% CI (0.89–1.41) (see Figure 2). The forest plot also displays the individual relative risks of every study with broad 95% CIs. According to the I^2^ statistics, no heterogeneity was observed (I^2^ = 0%).

## 4. Discussion

We reviewed the empirical studies on the risk of CNS progression in PVRL and compared the risk between bilateral and unilateral ocular involvement. We found no significant difference in CNS progression risk between bilateral and unilateral involvement in this meta-analysis. However, the results must be interpreted with caution since most of the studies were of moderate to high risk of bias. There was large variability between the studies in terms of CNS progression incidence and risk ratios. This was due to small sample sizes in the studies and differences in the definition and selection of the study samples.

Results presented in this review support findings from the previous literature regarding the risk of CNS disease development in patients with PVRL. The overall CNS progression incidence for patients with PVRL included in this review was 48%, which is within the range reported in the existing literature (33–60%) [4,5]. Follow-up periods varied widely between the included studies, with a range between one and 166 months and a median of 33 months. The percentage of patients with bilateral involvement reported in this review (68%) coincided with the percentage of the largest cohort in the existing literature [5].

In this systematic review, we made a distinction between CNS progression in bilateral and unilateral PVRL. This information has only limitedly been described in other reviews and retrospective studies regarding CNS progression of PVRL [4,5,62].

Our study inclusion was limited by the necessity for specific data for uni- versus bilateral ocular involvement. This led to the exclusion of some studies with potentially relevant information: Riemens et al. [11] described a cohort of 78 patients and stated that no difference in CNS progression between bilateral and unilateral PVRL was observed. The authors. supported this statement with a *p*-value of 0.94; no information on the ratio between bilateral and unilateral PVRL was given, and we were unable to retrieve more detailed information. Remarkably, Riemens et al. described that only 36% of the patients developed CNS progression. The authors explained this low percentage of CNS progression by the fact that patients with positive cerebrospinal fluid findings at initial staging were not considered primary vitreoretinal lymphoma. Additionally, patients by whom the diagnosis of PVRL was confirmed after CNS progression had developed were excluded. These two exclusion criteria are not followed by all studies. In our included studies, eight articles excluded patients with positive CSF cytology at the time of diagnosis, four articles did not exclude these patients, and thirteen articles did not state any information regarding CSF cytology at initial diagnosis. Furthermore, only twelve studies excluded patients by whom the diagnosis PVRL was confirmed after CNS progression had developed, five articles did not exclude these patients, and eight studies did not describe this. This could lead to a patient selection bias, which is taken into account in the risk of bias analysis. This possible selection bias could explain the difference in overall CNS progression between our included studies and the study of Riemens et al.

The two largest studies included in this review (Lam et al. [41] (*n* = 59) and Maruyama et al. [61] (*n* = 46)) also described no significant difference in CNS progression of unilateral versus bilateral ocular involvement. A note regarding the study of Lam et al. is that all 59 patients in this study received systemic chemotherapy, which is not the standard treatment approach. The authors described a CNS progression rate of 37% and did not exclude positive CSF cytology at the time of diagnosis with no information regarding the moment of inclusion.

Several studies have investigated different treatment strategies with different outcomes, but do not take into account if there is unilateral or bilateral initial involvement. Riemens et al. [11] described that different treatment strategies (only ocular therapy versus systemic therapy versus a combination of ocular and systemic therapy) did not result in differences in CNS progression rate, ocular relapse, and/or overall survival. Systemic therapy had a high adverse event rate. As stated before, no difference in laterality for CNS progression was described, although it is not known whether different treatment strategies were used for unilateral or bilateral PVRL. Different treatment strategies for unilateral or bilateral PVRL could potentially lead to differences in CNS progression. Lam et al. [41] described a cohort of 59 patients all receiving systemic chemotherapy, with higher overall survival rates and lower CNS progression than other large retrospective studies [4,11]. There was a high rate of ocular relapse, mainly since only a small percentage also received ocular therapy next to systemic therapy.

### 4.1. Strengths

For this systematic review, we followed a stringent protocol according to PRISMA guidelines and the QUIPS tool. In this way, we aimed to offer a systematic and critical appraisal of differences in CNS progression rates between bilateral and unilateral PVRL. Due to the systematic review of multiple studies, the data included here are representative of the general population since both western populations (USA and Europe, 52% of the included studies) and eastern populations (East Asia and India, 48%) are described.

### 4.2. Limitations

There were several limitations to this review, mostly due to the nature of the available original studies. The data of articles not published on PubMed or Embase may be lost, although the search strategy was very comprehensive. The review is based on mostly retrospective studies with small sample sizes, reducing confidence in the results. There is, to a large extent, a bias in terms of: (1) selection bias: in most studies, it is not described if the diagnosis of PVRL was confirmed in retrospect, i.e., after CNS progression; (2) confounding bias: the effects of possible differences in treatment protocols between studies, and between uni- and bilateral PVRL, is not accounted for in the analysis; (3) attrition bias: the studies had a large variation in follow-up period with a median ranging from seven to 68 months. Due to limited information, no time to event analysis could be calculated. Another major limitation in most published studies is the fact that no information is available on progression from unilateral to bilateral involvement, and afterwards CNS progression. Follow-up protocols for the screening of CNS progression were not described in most studies and only described globally in six studies [40,41,42,52,57,61].

Overall, quality assessment of the included studies by means of QUIPS demonstrated that most studies were at moderate to high risk of bias. Therefore, further research is still required to address the limitations discussed here with the goal of having a clear overview of the risk of CNS progression in PVRL with bilateral and unilateral involvement.

### 4.3. Implications

Knowledge of the relative risks of CNS progression for bilateral and unilateral PVRL could be used in designing better treatment and follow-up protocols. Over the past years, there have been many attempts to design individual treatments tailored to specific populations in order to increase the effectiveness of the treatment. For example, it is currently advised to use different treatment methods depending on the laterality of PVRL [63]. For patients with unilateral eye involvement, local intravitreal therapy should be considered, while systematic therapy should be considered for patients with bilateral involvement. Since our review yielded no difference in the risk of CNS progression between unilateral and bilateral PVRL, the rationale for this difference in treatment strategies is lacking. The current advice to use different protocols for follow-up or treatment depending on the laterality of PVRL should therefore be re-evaluated. Furthermore, this knowledge may support providing patients with more accurate information about the disease, treatment, and prognosis.

PVRL is a serious condition with high mortality rates. Yet, due to the rarity of the disease, systematic and large-scale studies are scarce. The present review adds to our understanding of this rare disease by providing a larger-scale synthesis of available data.

## 5. Conclusions

In this meta-analysis of available studies of patients with primary vitreoretinal lymphoma, the risk of developing CNS progression did not differ between patients with unilateral and bilateral ocular involvement. The available empirical studies are of limited quality, due to many limitations. Nevertheless, these preliminary results suggest that laterality of PVRL is not a valid factor for clinical risk stratification. Future studies should further collect and analyze data in order to increase confidence in these results, identify risk factors for CNS progression, and optimize clinical management.

## Figures and Tables

**Figure 1 cancers-14-02967-f001:**
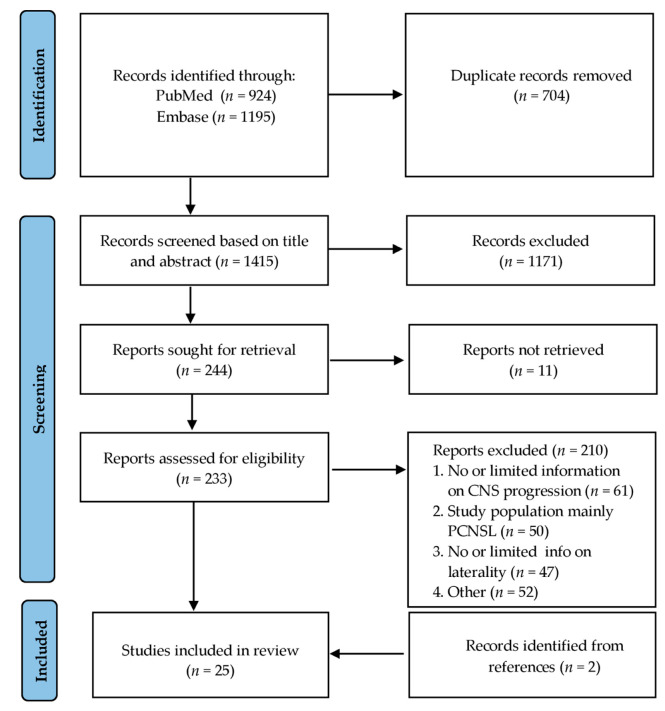
Flowchart illustrating study selection based on the PRISMA model.

**Figure 2 cancers-14-02967-f002:**
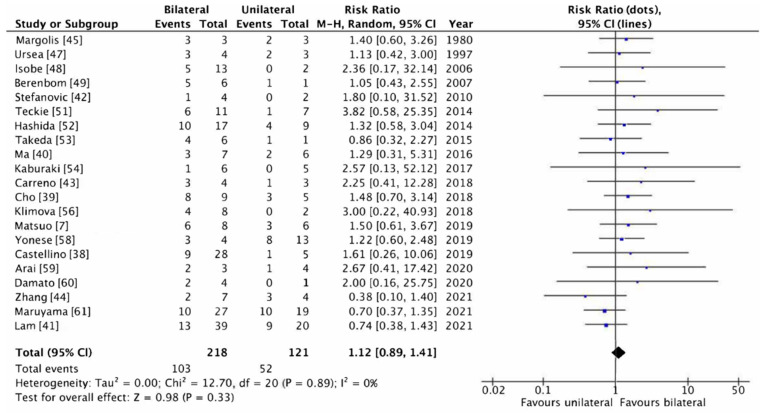
Forest plot of relative risk of CNS progression for PVRL with bilateral involvement compared to unilateral involvement.

**Table 1 cancers-14-02967-t001:** Study characteristics.

Study	Period (Years)	Study Design	Total Number of Patients with PVRL	CNS Progression for Patients with PVRL (%)	Laterality of PVRL	Time to CNS Progression: Mean (Range); in Months	Follow-Up Period: Median (Range); in Months	Initial Treatment Strategy
Margolis 1980 [45]	1968–1974	R	6	5/6 (83%)	3 B, 3 U	34 (14–54)	11 (5–39)	ORT, BRT, ChT, IT MTX
Soussain 1996 [46]	1992–1995	P	5	0/5 (0%)	5 B	n/a	21 (13–27)	ORT, BRT, ChT, ASCT
Ursea 1997 [47]	1997	R	7	5/7 (71%)	4 B, 3 U	Unknown (1–86)	39 (10–126)	Unknown
Isobe 2006 [48]	1990–2005	R	15	5/15 (33%)	13 B, 2 U	Unknown	19 (7–73)	BRT, ChT
Berenbom 2007 [49]	1995–2003	R	7	6/7 (86%)	6 B, 1 U	16 (Unknown)	12 (4–58)	BRT, ChT
Karma 2007 [50]	2000–2005	P	8	5/8 (63%)	8 B	Unknown	30 (10–49)	Unknown
Stefanovic 2010 [42]	2005	P	6	1/6 (17%)	4 B, 2 U	21 (n/a)	44 (10–51)	ORT, ChT
Teckie 2014 [51]	1999–2011	R	18	7/18 (39%)	11 B, 7 U	18 (2–42)	25 (2–150)	ORT, ChT
Hashida 2014 [52]	2001–2011	R	26	14/26 (54%)	17 B, 9 U	24 (8–65)	51 (27–81)	ChT, IT MTX, IVM, IVR
Takeda 2015 [53]	2008–2015	R	7	5/7 (71%)	6 B, 1 U	21 (4–48)	Unknown	Unknown
Ma 2016 [40]	2003–2013	R	13	5/13 (38%)	7 B, 6 U	Unknown	40 (4–123) *	ChT, IVM
Kaburaki 2017 [54]	2008–2015	P	11	1/11 (9%)	6 B, 5 U	9 (n/a)	49 (15–95) *	BRT, ChT, IVM
Mahajan 2017 [55]	2004–2015	R	7	5/7 (71%)	7 B	Unknown (4–36)	13 (6–64)	IVM
Carreno 2018 [43]	Unknown	R	7	4/7 (57%)	4 B, 3 U	15 (6–27)	7 (1–27)	Unknown
Cho 2018 [39]	2000–2014	R	14	11/14 (79%)	9 B, 5 U	17 (1–82)	Mean 39 (12–95)	BRT, ORT, ChT, IVM
Klimova 2018 [56]	2004–2016	R	10	4/10 (40%)	8 B, 2 U	34 (25–40)	53 (14–166)	ChT, IVM, BMT
DeLaFuente 2019 [57]	2005–2018	R	12	4/12 (33%)	12 B	Unknown	68 (17–154)	ORT, ChT, IVM
Matsuo 2019 [7]	2005–2019	R	14	9/14 (64%)	8 B, 6 U	15 (1–60)	31 (7–132)	BRT, ORT, ChT, ASCT, none
Yonese 2019 [58]	2007–2016	R	17	11/17 (63%)	4 B, 13 U	29 (11–67)	33 (11–103)	ChT, IVM
Castellino 2019 [38]	1990–2018	R	33	10/33 (30%)	28 B, 5 U	Unknown	36 (Unknown)	BRT, ORT, ChT, ASCT, IVM, IVR
Arai 2020 [59]	2011–2018	R	7	3/7 (43%)	3 B, 4 U	18 (11–24)	36 (21–67)	ChT, IVM
Damato 2020 [60]	2013–2018	P	5	2/5 (40%)	4 B, 1 U	8 (4 –13)	44 (30–50)	ChT, sR
Maruyama 2021 [61]	2004–2020	R	46	20/46 (43%)	27 B, 19 U	22 (1–55)	Unknown	BRT, Cht, IT MTX, IVM, IVR, sR, none
Zhang 2021 [44]	2018–2020	P	11	5/11 (45%)	7 B, 4 U	9 (1–25)	18 (11–28)	ChT, IVM, sR, lenalidomide
Lam 2021 [41]	2011–2018	R	59	22/59 (37%)	39 B, 20 U	Unknown	61 (Unknown)	ORT, ChT, ASCT, IVM, sR

* Follow-up period of combined group of PCNSL and PVRL, not stated specifically regarding PVRL. Abbreviations: PVRL = primary vitreoretinal lymphoma; CNS = central nervous system; R = retrospective; P = prospective; B = bilateral; U = unilateral; ORT = ocular radiation therapy; BRT = brain radiation therapy; ChT = systemic chemotherapy; IT MTX = intrathecal methotrexate; ASCT = autologous stem cell transplantation; IVM = intravitreal methotrexate; IVR = intravitreal rituximab; sR = systemic rituximab.

**Table 2 cancers-14-02967-t002:** Quality assessment of data using QUIPS.

Risk of Bias							
Study	Study Participation	Study Attrition	Prognostic Factor	Outcome	Study Confounding	Statistical Analysis and Reporting	Overall
Margolis [45]	** ● **	** ● **	** ● **	** ● **	** ● **	** ● **	** ● **
Soussain [46]	** ● **	** ● **	** ● **	** ● **	** ● **	** ● **	** ● **
Ursea [47]	** ● **	** ● **	** ● **	** ● **	** ● **	** ● **	** ● **
Isobe [48]	** ● **	** ● **	** ● **	** ● **	** ● **	** ● **	** ● **
Berenbom [49]	** ● **	** ● **	** ● **	** ● **	** ● **	** ● **	** ● **
Karma [50]	** ● **	** ● **	** ● **	** ● **	** ● **	** ● **	** ● **
Stefanovic [42]	** ● **	** ● **	** ● **	** ● **	** ● **	** ● **	** ● **
Teckie [51]	** ● **	** ● **	** ● **	** ● **	** ● **	** ● **	** ● **
Hashida [52]	** ● **	** ● **	** ● **	** ● **	** ● **	** ● **	** ● **
Takeda [53]	** ● **	** ● **	** ● **	** ● **	** ● **	** ● **	** ● **
Ma [40]	** ● **	** ● **	** ● **	** ● **	** ● **	** ● **	** ● **
Kaburaki [54]	** ● **	** ● **	** ● **	** ● **	** ● **	** ● **	** ● **
Mahajan [55]	** ● **	** ● **	** ● **	** ● **	** ● **	** ● **	** ● **
Carreno [43]	** ● **	** ● **	** ● **	** ● **	** ● **	** ● **	** ● **
Cho [39]	** ● **	** ● **	** ● **	** ● **	** ● **	** ● **	** ● **
Klimova [56]	** ● **	** ● **	** ● **	** ● **	** ● **	** ● **	** ● **
DeLaFuente [57]	** ● **	** ● **	** ● **	** ● **	** ● **	** ● **	** ● **
Matsuo [7]	** ● **	** ● **	** ● **	** ● **	** ● **	** ● **	** ● **
Yonese [58]	** ● **	** ● **	** ● **	** ● **	** ● **	** ● **	** ● **
Castellino [38]	** ● **	** ● **	** ● **	** ● **	** ● **	** ● **	** ● **
Arai [59]	** ● **	** ● **	** ● **	** ● **	** ● **	** ● **	** ● **
Damato [60]	** ● **	** ● **	** ● **	** ● **	** ● **	** ● **	** ● **
Maruyama [61]	** ● **	** ● **	** ● **	** ● **	** ● **	** ● **	** ● **
Zhang [44]	** ● **	** ● **	** ● **	** ● **	** ● **	** ● **	** ● **
Lam [41]	** ● **	** ● **	** ● **	** ● **	** ● **	** ● **	** ● **

**● **= high risk of bias; **●**= moderate risk of bias; **● **= low risk of bias.

**Table 3 cancers-14-02967-t003:** Relative risk of CNS progression for PVRL with bilateral involvement compared to PVRL with unilateral involvement. RR > 1 suggests higher risk of CNS progression in cases with bilateral involvement; RR < 1 suggests higher risk for unilateral cases.

Study	Bilateral–CNS Progression	Bilateral–No CNS Progression	Unilateral–CNS Progression	Unilateral–No CNS Progression	Relative Risk	95% CI
Margolis [45]	3/3 (100%)	0/3 (0%)	2/3 (67%)	1/3 (33%)	1.40	[0.60, 3.26]
Soussain [46]	0/5 (0%)	5/5 (100%)	-	-	-	-
Ursea [47]	3/4 (75%)	1/4 (25%)	2/3 (67%)	1/3 (33%)	1.13	[0.42, 3.00]
Isobe [48]	5/13 (38%)	8/13 (62%)	0/2 (0%)	2/2 (100%)	2.36	[0.17, 32.14]
Berenbom [49]	5/6 (83%)	1/6 (17%)	1/1 (100%)	0/1 (0%)	1.05	[0.43, 2.55]
Karma [50]	5/8 (63%)	3/8 (38%)	-	-	-	-
Stefanovic [42]	1/4 (25%)	3/4 (75%)	0/2 (0%)	2/2 (100%)	1.80	[0.10, 31.52]
Teckie [51]	6/11 (55%)	5/11 (45%)	1/7 (14%)	6/7 (86%)	3.82	[0.58, 25.35]
Hashida [52]	10/17 (59%)	7/17 (41%)	4/9 (44%)	5/9 (56%)	1.32	[0.58, 3.04]
Takeda [53]	4/6 (67%)	2/6 (33%)	1/1 (100%)	0/1 (0%)	0.86	[0.32, 2.27]
Ma [40]	3/7 (43%)	4/7 (57%)	2/6 (33%)	4/6 (67%)	1.29	[0.31, 5.31]
Kaburaki [54]	1/6 (17%)	5/6 (83%)	0/5 (0%)	5/5 (100%)	2.57	[0.13, 52,12]
Mahajan [55]	5/7 (71%)	2/7 (29%)	-	-	-	-
Carreno [43]	3/4 (75%)	1/4 (25%)	1/3 (33%)	2/3 (67%)	2.25	[0.41, 12.28]
Cho [39]	8/9 (89%)	1/9 (11%)	3/5 (60%)	2/5 (40%)	1.48	[0.70, 3.14]
Klimova [56]	4/8 (50%)	4/8 (50%)	0/2 (0%)	2/2 (100%)	3.00	[0.22, 40,93]
DeLaFuente [57]	4/12 (33%)	8/12 (67%)	-	-	-	-
Matsuo [7]	6/8 (75%)	2/8 (25%)	3/6 (50%)	3/6 (50%)	1.50	[0.61, 3.67]
Yonese [58]	3/4 (75%)	1/4 (25%)	8/13 (62%)	5/13 (38%)	1.22	[0.60, 2.48]
Castellino [38]	9/28 (32%)	19/28 (68%)	1/5 (20%)	4/5 (80%)	1.61	[0.26, 10.06]
Arai [59]	2/3 (67%)	1/3 (33%)	1/4 (25%)	3/4 (75%)	2.67	[0.41, 17.42]
Damato [60]	2/4 (50%)	2/4 (50%)	0/1 (0%)	1/1 (100%)	2.00	[0.16, 25,75]
Maruyama [61]	10/27 (37%)	17/27 (63%)	10/19 (53%)	9/19 (47%)	0.70	[0.37, 1.35]
Zhang [44]	2/7 (29%)	5/7 (71%)	3/4 (75%)	1/4 (25%)	0.38	[0.10, 1.40]
Lam [41]	13/39 (33%)	26/39 (67%)	9/20 (45%)	11/20 (55%)	0.74	[0.38, 1,43]

Abbreviations: CNS: central nervous system; CI: confidence interval.

## Data Availability

All data, as extracted from the original reports, can be found in the published tables and text.

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
