# Peer review of "Central Nervous System Progression in Primary Vitreoretinal Lymphoma with Bilateral and Unilateral Involvement: A Systematic Review and Meta-Analysis"

_cancers, 2022, doi:10.3390/cancers14122967_

Round 1
Reviewer 1 Report
This systematic analysis is a laborious effort in order to provide clinicians confronted with primary vitreoretinal lymphoma / primary intraocular lymphoma (PIOL) with prognostic factor(s) influencing possible progress to primary CNS lymphoma (PCNSL), which frequently occurs in this condition.
No standard of Treatment is established in PIOL, which may occur unilaterally or bilaterally and often is accompanied (either at initial presentation or during the course of the disease) with PCNSL. PIOL/PCNSL most likely presents a systemic disease and may make systemic therapy necessary, such that some experts even suggest HDASCT. However, that this might be superior to local Tumor Control in the eye(s) alone by focal therapy has not been shown.
To weigh the scientific merit of this manuscript it is important to consider, that PCNSL is accompanied by PIOL in 15-25%, that PIOL will eventually present as PCNSL in > 50% and that initial unilateral disease will frequently present as bilateral eye involvement in up to 90% during follow-up. That means, clinicians have to carefully monitor the brain and the eyes during follow-up after initial presentation and completion of therapy, no matter, if the CNS has been involved first or the eyes (bilaterally or unilaterally). This Consensus on monitoring and Treatment planning could be modified only by extreme strong prognostic factors. And the Authors undertook an effort to identify one.
Unfortunately the present paper cannot provide a strong predictor. The meta-analysis has been carried out in a decent way, the methodology is sound, the selection of papers is appropriate. However, the source of Information, i.e. the Quality of data going into the Analysis is not. We would need a carefully documented series of patients with homogeneous therapy and standardized follow up with regular Monitoring by cMRI, ophthalmologic investigation at set time points, ideally followed prospectively. Now we are left with the Observation, that probably there is no difference in the risk of developing CNS disease between unilateral or bilateral PIOL, but the results are far from being able to prove this.
Reviewer 2 Report
Well written comprehensive analysis of the topic. Minor comments
Line 79: evolvement should be involvement.
Table 3 is cluttery The second and fourth columns do not add anything. Would leave just the 2 columns Bilat – CNS progression and Unilat – CNS progression.
Reviewer 3 Report
The authors add a valuable systematic review and meta-analysis on the important topic, whether there is a higher incidence for PCNSL after bilateral PVRL in comparison to unilateral PVRL. The manuscript is well written and uses appropiate methods to do the analysis.
Some alterations have to be done before publication.
Abstract: State with the last sentence of the backround the hypothesis of this analysis starting with "Objective/Aim of this analysis was...". Add the information for the analysis in the M&M section (PRISMA; Grading tool, statistical analysis)
Introduction: The part is a review in the review. A lot of information on PVRL is not necessary to introduce into the topic. Therefore, alter this chapter by straight going the topic and introducing just the topic and remove all other information (The need on information, whether there is a higher risk to develop PCNSL if there is bilateral PVRL manifestation. And what implications this might have for the daily clinical care of PVRL.
Material&Methods as well as Results are excellent and clearly performed.
Discussion: Add to the limitation, that patients with PVRL (either unilateral or bilateral) have been treated different (w or w/o systemic treatment), which might have an impact on the analysed data. Also add in the implication some criteria suggestions for performing better studies on this subject in the future.
Round 2
Reviewer 1 Report
/